# "My Pronouns Are": Pronoun-face mismatch performance and self-report attitudes to gender categorizaton across generations

Melissa Allen‡, Catherine Hobbs [ID]‡*, Will Mills [ID], Stephen Hinde, Bruce Hood

University of Bristol, Bristol, United Kingdom

‡ These authors are joint first authors on this work.
* c.hobbs@bristol.ac.uk

## Abstract

Gender attribution based on face categorization may undergo age-related changes. Specifically, older individuals with more exposure to faces may have stronger stereotypical representations which leads to more automatic categorization. Investigating automatic binary pronoun assignment (he/she) allows us to examine whether older adults show greater categorical interference, with slower responses when pronouns do not match stereotypical faces. We tested this proposal by presenting individuals (N = 600: age range = 18–69 yrs) with a pronoun–face congruence task contrasting stereotypical faces with either congruent or incongruent binary (he/she) pronouns. Individuals also reported their explicit attitudes to preferred gender pronoun use with older individuals expressing more negative attitudes towards pronoun categorization as well as greater difficulty, but fewer worries than younger participants. Although older participants were overall slower, importantly, there were no age differences in pronoun–face interference. These findings support an age difference in explicit attitudes towards gender pronoun categorization that are not mirrored in reaction time measures.

## Introduction

Gender identity is currently a major social concern. One specific contentious issue is *misgendering* which is the incorrect use of a pronoun to refer to an individual who does not identify with a gender label that is applied to them. It is particularly relevant to transgender, non-binary and gender-fluid communities [1] but also cis-gender individuals who identify with their assigned sex at birth but are misperceived as belonging to a different category. There are also generational differences in how this issue of diversity is regarded, leading to different attitudes towards gender categorization using pronouns. For example, Risman [2] highlights that in Western societies, millennials can reject the notion that sex and gender need to be correlated, with some individuals wholly rejecting gender as a binary construct. A recent review of attitudes

**Data availability statement:** Yes - all data are fully available without restriction; Data are available at the University of Bristol data repository (data.bris) at https://doi.org/10.5523/bris.2qw098ihp7rdp2k2un82zpsgec.

**Funding:** The author(s) received no specific funding for this work.

**Competing interests:** The authors have declared that no competing interests exist.

to using gendered language concluded that younger people may adapt more easily to new forms than older people, who have been exposed to traditional language structures for a longer time [3].

Related to exposure, another reason for misgendering may be that coding for sex becomes automatic and subject to interference [4]. The classic demonstration of this is the *Who Said What?* paradigm using a memory confusion protocol, where participants were asked to remember a series of interactions between individuals engaged in conversation [5]. They were then presented with statements made during the conversation paired with photographs of the conversationalists and asked to identify who said what. Misattributions revealed that participants more readily confused individuals whom they have encoded as members of the same sex category than those whom they categorized as members of the opposite sex category.

Interference can occur when individuals are faced with incongruent information or responses at the same time. For example, when identifying the words "man" versus "girl" spoken by either a male or female voice, participants were faster when the target and voice were congruent (i.e., male voice saying "man") [6]. Later studies had participants judge whether spoken gender stereotypical names and words (e.g., female: "Susan, lipstick, pink" male: "Peter, soldier, tough") matched the speaker's voice who was either male or female [7]. Both adults and children had longer response times when a voice's sex was incongruent with a name or spoken word, confirming that gendered words prime voice categorization. Other more recent studies using gendered prime words, as typically found in Spanish and Italian, found both behavioural and physiological congruence effects for both pronouns and faces [8,9]. These studies form the basis for the current study where we wanted to investigate whether there was an interference effect when participants were presented with typical male and female faces that were either matched or mismatched with pronouns ("he" or "she"), and whether this interference changes with age.

Interference occurs due to *both* the automatic processing of dominant stimulus dimensions [10], as well as a lack of inhibitory control to suppress these automatic responses that changes with age [11]. As individuals gain experience with certain types of faces, especially those encountered more frequently, facial processing becomes more efficient but also more entrenched [12]. One of the best examples of this is the other-race effect where face recognition is facilitated for own race faces but impaired for other race faces which has been shown to increase with exposure [13]. The automaticity of gender categorization based on facial features, when coupled with age-related declines in response inhibition, may therefore lead to increased instances of misgendering. In the current study we predicted a robust pronoun-face congruence effect (based on pilot work) but were agnostic as to whether this interference was primarily due to stronger pronoun-face schemas or weaker inhibition. Furthermore, our question of interest was whether or not the interference would change with age.

If a pronoun-face matching task elicits an interference effect, is it possible that this is related to attitudes and self-reports? Response times are often interpreted as measures of implicit processing but are problematic in how they are interpreted in

relation to explicit measures [14]. For example, another measure of implicit processing is the Implicit Attitudes Test (IAT; [15]) which has been used to demonstrate automatic categorization and biases in other person dimensions such as age [16] and race [17]. Although use and application of the IAT has been extensive and influential, it is controversial. The term "implicit attitude" is arguably a contradiction because attitude usually means a stable explicit evaluation or belief held by a person whereas implicit suggests something unconscious, automatic, or unintentional. Moreover, there is little construct validity in that implicit measures on the IAT do not always correlate strongly with explicit measures and there is also wide variation depending on the category being judged.

As there were no validated questionnaires on self-report attitudes to pronoun categorization, we constructed and validated one to provide a measure of explicit attitudes towards preferred pronoun categorization [see S1 Appendix]. In addition to this broad attitudinal scale, we also used two related but conceptually distinct self-report measures assessing perceived difficulties using preferred pronouns and worries about using pronouns incorrectly. Difficulties reflect perceived cognitive or linguistic challenges in applying preferred pronouns (e.g., remembering or consistently using them), whereas worries capture affective concerns about misgendering and its potential social consequences. Although these constructs were assessed using separate scales and were empirically distinguishable, they are conceptually related in that all reflect how individuals evaluate, engage with, and emotionally respond to changing social norms around gender categorization. While we predicted that there will be age-related differences on pronoun-face interference in our study, we did not predict that there will be a strong association with explicit attitudes to gender pronouns and pronoun categorization (as observed with the IAT). In other words, participants may not be aware that their attitude towards use of gender pronouns, or any difficulties or worries they experience in relation to gender categorization are mirrored by implicit processes.

In this study we sought to investigate age-related changes in speeded gender pronoun categorization based on facial appearance and whether this relates to explicit attitudes towards gender pronoun categorization in general, namely beliefs regarding normative patterns of pronoun assignment. We define gender pronoun categorization as the process of grouping, selecting, and applying personal pronouns (e.g., *he/him, she/her, they/them*) on the basis of gender-related categories, which may be assigned, self-identified, context-dependent, or institutionally specified. We acknowledge that such normative patterns vary historically and culturally. For example, gender fluid cultures that have long existed in non-Western and Indigenous societies [18], such as the Hijra, Muxe, Bissu priests, Calalai and Mahu, among others [19].

In our paradigm, we used images of binary and stereotypically presenting male or female individuals though we acknowledge that this does not represent the full spectrum of gender and identity or allow us to test specific interference effects beyond binary presentations. Following initial investigations with a pilot study, we examined 600 participants aged from 18 to 69 years to determine whether there was any difference in interference effects over age in two conditions of a pronoun–face congruence paradigm. In one condition, pronoun assignment was either congruent or incongruent with typical gender categorization based on facial appearance. To check whether any age differences on the task were specific to pronoun categorization, we also ran a control condition where an arrow (up versus down) was either congruent or incongruent with orientation of the face.

We hypothesised that older participants would show a stronger pronoun–face interference effect, as measured by longer reaction times for incongruent versus congruent trials, for the gender pronoun relative to the face orientation control task, compared to younger participants (Hypothesis 1). We also hypothesised that as participants got older, they would report a more negative attitude towards gender pronoun categorization and greater difficulties about using preferred gender pronouns as measured by a self-report questionnaire (Hypothesis 2). Finally, in line with our pilot studies and work showing weak or absent relationships between measures assessed using response-conflict paradigms and explicit self-reports, we hypothesised that explicit attitudes would not be associated with performance on the pronoun–face congruence task (Hypothesis 3). As age may capture both life-course variation and generational differences in social experience, we examined both continuous age and generational groups in our analyses.

## Method

### Ethics

This research was approved by the University of Bristol School of Psychological Science Research Ethics Board (14493). Participants provided written informed consent electronically by indicating consent via checkbox selection.

### Pilot study

We ran an initial online pilot study of 103 English-speaking UK-based adults (18–67 years) to examine age-related performance on the pronoun–face task and we also measured participants' explicit attitudes to preferred gender pronouns with a questionnaire (see S1 Appendix for results). We found no evidence of a relationship between interference and explicit attitudes but explicit attitudes towards gender pronouns were generally more negative with age. However, this initial study was limited by a small sample size powered to detect only a medium effect. We were therefore underpowered to detect smaller, more subtle, age-related differences. Furthermore, the self-report measure we used to measure attitudes to gender pronouns was brief and not validated. Although this initial study was flawed, we were able to use it to calculate the pre-requisite number of participants to provide a robust test for an age-related interference effect related to mismatched pronouns and typical faces and we developed a new validated questionnaire in consultation with experts in gender studies.

### Current study

The study hypotheses, methods, and statistical analyses were pre-registered on Open Science Framework (https://osf.io/kqsfe).

**Participants.** We conducted an a priori sample size calculation using the means, standard deviations, and within-subject correlation from the initial pilot study as a starting point to test for an interaction effect of age, task, and congruency for mean reaction times (ms) as outlined in Hypothesis 1. We simulated data with the hypothesised pattern (S1 Fig in S1 Appendix) to generate a power calculation based on repeated sampling. Based on 5000 iterations we estimated that 600 participants would provide 86.5% power (95% CI: 85.5%, 87.4%) to detect our hypothesised effect at a p-value of 0.05.

Participants were recruited via the online platform 'Prolific' between the 18th-28th November 2024. Participants were required to be located in the United Kingdom, have normal or corrected-to-normal vision and speak English fluently. To ensure we had a uniform distribution of age we recruited four groups of participants according to generation (aged 18–27 years, 28–43 years, 44–59 years, 60–69 years). To promote data quality, we restricted participants to those who had completed ≥ 5 Prolific studies with a 100% acceptance rate.

Participants completed a qualification task at the start of the study procedure to ensure that they understood the task instructions before moving to the main tasks. Participants were required to response correctly on at least 16/20 trials before they could proceed to the main study and were excluded after three failed attempts.

Of the 743 participants who consented to take part in the study, 682 (91.8%) participants completed the study procedure (40 failed the qualification task, 6 failed ≥ 2 attention checks, 15 withdrew). We excluded 82 participants from the analysis (26 had < 70% accuracy on ≥ 1 of the tasks, 4 had ≥ 10% invalid trials on ≥ 1 of the pronoun–face congruence tasks, 4 experienced technical problems resulting in duplicate completion of a task, and 48 failed an attention check), leaving a total sample of 600 participants (M = 43.7 years; range = 18–69 years; 48.2% female, 51.2% male, 0.6% non-binary/other).

**Procedure.** Participants completed the study procedure remotely using Gorilla [20], a platform for online behavioural experiments. After providing informed consent, participants completed a brief screening questionnaire to verify recruitment information. Participants were then asked to complete a qualification task which followed the same procedure as the

main tasks but used non-face related stimuli. Those that passed the qualification task (correct responses in ≥ 16/20 trials) in less than three attempts proceeded to the main study procedure, where they completed a the pronoun-face and orientation-face tasks, in a randomised order. In the pronoun-face task, participants were presented with an image of a man or woman along with a binary pronoun ('he' or 'she'). Participants were asked to indicate if the pronoun matches or mismatches with the visual presentation of the individual's gender in the image using the computer keyboard. In the orientation-face task, participants were shown the same images of a man or a woman displayed either upright or inverted alongside text stating 'upright' or 'upside down'. Participants were asked to indicate whether the text matches the orientation of the image using the computer keyboard. The purpose of the orientation-face task was to provide a simple control to test automatic interference effects that were not gender specific. Across both tasks a time limit response of 3 seconds was imposed. Participants completed two blocks of 48 trials per task. Key assignment was randomised; half of participants were instructed to use the left key to indicate a match and the right key to indicate a mismatch, and the other half used the reverse.

After completing the orientation- and pronoun-face tasks, participants completed the study questionnaires (see Materials). We included attention checks interspersed through the study procedure in the form of four statements explicitly instructing participants to select a specific response (e.g., 'Please select 'Strongly Disagree''). At the end of the study procedure participants provided final informed consent.

**Materials.** For the Orientation- and Pronoun-Face Congruence Tasks, stimuli were adapted from the Face Research London Dataset [21] and consisted of 48 pictures of individual adults who visually presented as either male or female. Actors wore neutral coloured clothing, presented with neutral facial expressions, and faces were presented against an identical background across the stimuli set. The face stimuli are standardized and have been used previously to test stereotypes [22] and gender biases [23].

In the initial study, we used a set of questions to measure explicit attitudes to using gender pronouns, but these were limited in that they were ad hoc and had not been validated using external experts and statistical analysis. Therefore, in the current study, we developed a validated questionnaire about overall attitudes to gender pronouns using an extensive procedure including review by academic experts and a lay audience, factor analysis, and establishment of convergent and test-retest reliability (See S1 Appendix for an overview). The items captured related facets of pronoun attitudes, including openness to non-binary pronouns, comfort and willingness to use preferred pronouns, respect for individual identity, and endorsement or rejection of traditional binary beliefs. Exploratory and confirmatory factor analyses in independent samples supported a single-factor solution, indicating that these related facets converged on a single underlying attitudinal construct (S1 Appendix).

Participants indicated their agreement relating to 8 statements about gender pronouns using a 5-point Likert scale from Strongly Disagree (1) to Strongly Agree (5) with two items reverse-coded (RC). We calculated a sum score to indicate overall explicit attitudes towards gender pronouns where higher scores indicate a greater positive attitude. We also included two questions about difficulty using gender pronouns and three about worries using gender pronouns (see Table 1). These were not included in the overall attitudes questionnaire because factor analysis indicated that the loading with other items were low. Finally, to ensure potential effects for worries about using gender pronouns were not confounded by general levels of anxiety we measured anxiety using the GAD-7 [24]. The GAD-7 consists of 7-items relating to experience of generalised anxiety symptoms in the previous two weeks.

## Statistical analysis

As outlined in our pre-registration we excluded participants who failed ≥ 1 attention check, participants who did not complete the qualification task with ≥ 16 accurate trials after 3 attempts, and participants with < 70% accuracy on ≥ 1 of the orientation or pronoun–face tasks. In addition to these pre-registered criteria, to ensure data quality we also excluded participants with ≥ 10% invalid trials on ≥ 1 of the orientation or pronoun–face tasks and those who experienced technical

**Table 1. Attitude, difficulty and worry related to using gender pronouns.**

*Attitudes to Gender Pronouns*

1) I think that traditional gender pronouns are outdated and do not reflect everyone's experiences.

2) I feel comfortable changing my language to use someone's gender pronouns.

3) I would feel guilty if I used the wrong gender pronouns.

4) I believe that men have penises and women have vaginas, and that this defines their gender pronouns (RC).

5) I think that using preferred gender pronouns shows respect.

6) I think that society is too focused on accommodating gender identities (RC).

7) I feel comfortable with others stating their gender pronouns.

8) I think that it's important to use the gender pronouns someone prefers, even if they don't match how I think they look.

*Difficulties using Gender Pronouns*

1) Overall, I find it easy to use someone's gender pronouns.

2) I find it easier to use the right gender pronouns for someone when their appearance matches the usual way people with those gender pronouns look.

*Worries about using Gender Pronouns*

1) I worry about being judged or criticised if I use the wrong gender pronouns.

2) I worry about asking someone for their preferred gender pronouns because I might get it wrong.

3) I worry about accidentally using the wrong gender pronouns.

problems resulting in duplicate completion of the orientation or pronoun–face tasks. We calculated mean reaction times in ms per participant, condition, and congruency to use as our main outcome measure. We excluded invalid responses (< 200 ms, > 3000 ms) and inaccurate trials in this calculation.

To test Hypothesis 1, we conducted a mixed-effects linear regression model with mean reaction times as a continuous outcome. Fixed effects included age (in years), task type, congruency, and all interaction terms between age, task, and congruency. Participant was included as a random intercept to account for within-subject variability. To complement this analysis conducted at a mean RT level, we also modelled the data at the trial level to capture stimuli-level variance and within-participant fluctuations in reaction times (See S1 Appendix). We also explored potential non-linear relationships between age and mean reaction times based on visual examination of the data.

For Hypothesis 2, we used a linear regression model with total scores on the Explicit Attitudes to Gender Pronouns questionnaire as a continuous outcome, and age in years as a predictor. As exploratory analyses, we also repeated these models with sum scores for self-report questions about difficulties and worries using gender pronouns as predictors. Evaluation of model assumptions showed that, although the central distributions were approximately normal, the difficulties and worries scales, comprising of only two and three items respectively, were discrete and bounded, resulting in minor deviations in the tails of the distributions. To account for this, all linear models for Hypothesis 2 were estimated with heteroskedasticity-consistent (HC3) robust standard errors. In addition, proportional-odds ordinal models were conducted as sensitivity analyses to confirm the robustness of the findings (reported in S1 Appendix).

For Hypothesis 3, we calculated the difference in reaction times (ms) between incongruent and congruent trials in the pronoun–face congruence task (the interference effect) and used this as a continuous outcome in a linear regression model. We entered age in years, and sum scores for each set of questions as predictors independently and in interaction in individual models.

We also investigated whether changes in processing of gender pronouns were best characterised by continuous changes across the life course or differences in experiences unique to generations by repeating our analyses for Hypotheses 1 and 2 with generation (Gen Z, Millennial, Gen X, Boomer) as a predictor in place of age. As age and generation are

confounded in cross-sectional designs, these analyses cannot fully separate aging-related processes from cohort effects but provide complementary perspectives on how age relates to performance and attitudes. We evaluated model fit using the Akaike Information Criterion (AIC), Bayesian Information Criterion (BIC), and where appropriate likelihood ratio tests (LRT). As results were consistent when using age and generation as predictors we report these results as supplementary (S6–S9 Tables in S1 Appendix).

## Results

### Orientation and pronoun–face congruence task performance

Participants showed a high level of accuracy on the tasks indicating that they understood the task instructions (Orientation Mean 94.5%, SD 4.4%; Pronoun-Face Mean 94.9%, SD 4.2%). Mean reaction times were slower for incongruent versus congruent trials for both the orientation (Congruent: Mean 1073.72 SD 212.28, Incongruent: Mean 1184.26 SD 232.1) and pronoun-face (Congruent: Mean 1019.91 SD 196.66, Incongruent: Mean 1113.75 SD 216.21) tasks, reflecting the expected interference effect whereby incongruent stimuli are more cognitively demanding.

### Hypothesis 1: We hypothesise that older participants will show a stronger interference effect, as measured by longer reaction times for incongruent versus congruent trials, for the pronoun-face task relative to the face orientation task compared to younger participants

We found evidence of main effects for age, task, and congruency (Fig 1; Table 2). Older participants were on average slower; for every additional year of age participants on average became 4.62 ms slower (95% CI: 3.58, 5.66, p < .001). Additionally, participants were on average 47.86 ms faster on the pronoun-face versus face orientation task (95% CI: −80.02, −15.69, p = 0.004). Participants were on average 96.11 ms slower for incongruent versus congruent trials, reflecting expected interference effects (95% CI: 63.95, 128.28, p < .001).

However, we did not find evidence to support our hypothesis of interaction effects between congruency, task, and age. Participants were slower for incongruent versus congruent trials in both the face orientation and pronoun-face tasks (task x Congruency: b = −7.96, 95% CI: −53.42, 37.49, p = 0.731). Additionally, older participants tended to be slower

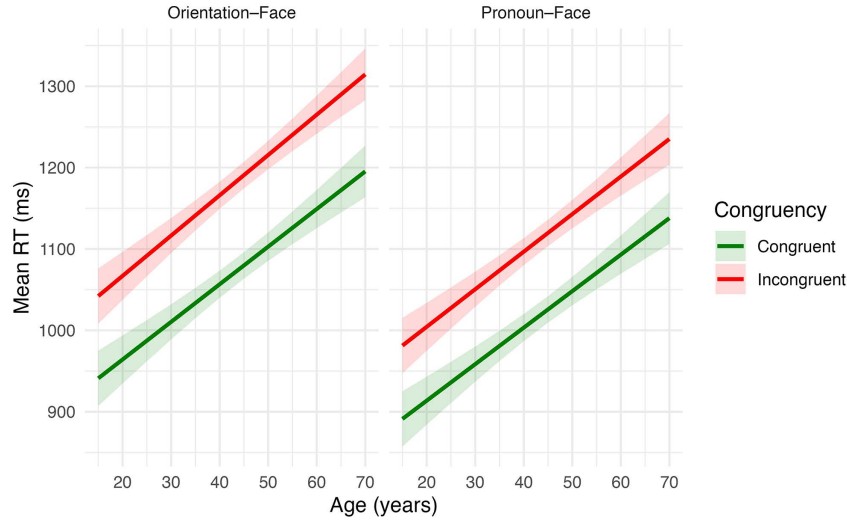

**Fig 1. Results from mixed-effects linear regression models for Hypothesis 1 showing predicted mean reaction times (ms) in relation to age (years) by task and congruency of trials.** Shaded areas represent 95% confidence intervals.

**Table 2. Results from mixed-effect linear regression models predicting mean reaction times (ms) from age (years), task, and congruency.**

| Predictor | b | Lower CI | Upper CI | p |
|---|---|---|---|---|
| Intercept | 871.72 | 823.53 | 919.91 | <.001 |
| Congruency | | | | |
| Congruent (Reference) | | | | |
| Incongruent | 96.11 | 63.97 | 128.26 | <.001 |
| Task | | | | |
| Orientation (Reference) | | | | |
| Pronoun-Face | −47.86 | −80.00 | −15.71 | 0.004 |
| Age | 4.62 | 3.58 | 5.66 | <.001 |
| Congruency x Task | −7.96 | −53.42 | 37.49 | 0.731 |
| Congruency x Age | 0.33 | −0.36 | 1.02 | 0.350 |
| Task x Age | −0.14 | −0.83 | 0.56 | 0.700 |
| Congruency x Task x Age | −0.20 | −1.18 | 0.78 | 0.689 |

irrespective of congruency (Age x Congruency: $b = 0.33$, 95% CI: −0.36, 1.02, $p = 0.350$) or task (Age x task: $b = -0.14$, 95% CI: 0.83, 0.56, $p = 0.700$). Finally, there was no significant three-way interaction between age, congruency, and task; participants showed a similar magnitude of interference effects for the face orientation and pronoun-face task irrespective of their age (Age x Task x Congruency: $b = -0.20$, 95% CI: −1.18, 0.78, $p = 0.689$). Our findings were consistent when modelled at a trial-level (S4 Table in S1 Appendix).

We did not find evidence of a non-linear association between age and task performance (S5 Table in S1 Appendix).

## Hypothesis 2: As participants get older, they will report a more negative attitude towards gender pronouns and greater difficulties using gender pronouns as measured by self-report questionnaires

In support of Hypothesis 2, older participants had a more negative attitude to gender pronouns as indicated by lower scores on the attitudes to gender pronouns questionnaire ($b = -0.11$, 95% CI: −0.15, −0.07, $p < .001$; Table 3; Fig 2). Older participants

**Table 3. Results from linear regression models predicting self-reported attitudes toward gender pronouns (overall attitudes, perceived difficulties, and worries) from age (years).**

| Predictor | b | Lower CI | Upper CI | p-value |
|---|---|---|---|---|
| *Attitudes to Gender Pronouns* | | | | |
| Intercept | 30.06 | 28.14 | 31.99 | <.001 |
| Age | −0.11 | −0.15 | −0.07 | <.001 |
| *Difficulties using Gender Pronouns* | | | | |
| Intercept | 8.22 | 7.90 | 8.55 | <.001 |
| Age | −0.01 | −0.02 | −0.01 | <.001 |
| *Worries about using Gender Pronouns* | | | | |
| Intercept | 10.65 | 9.84 | 11.47 | <.001 |
| Age | −0.04 | −0.06 | −0.02 | <.001 |
| *Adjusting for generalised anxiety* | | | | |
| Intercept | 9.58 | 8.64 | 10.51 | <.001 |
| Age | −0.03 | −0.05 | −0.01 | 0.002 |
| GAD-7 Sum Score | 0.14 | 0.08 | 0.20 | <.001 |

Note: All models use heteroskedasticity-consistent (HC3) robust standard errors; CIs and p-values are based on the robust variance estimates.

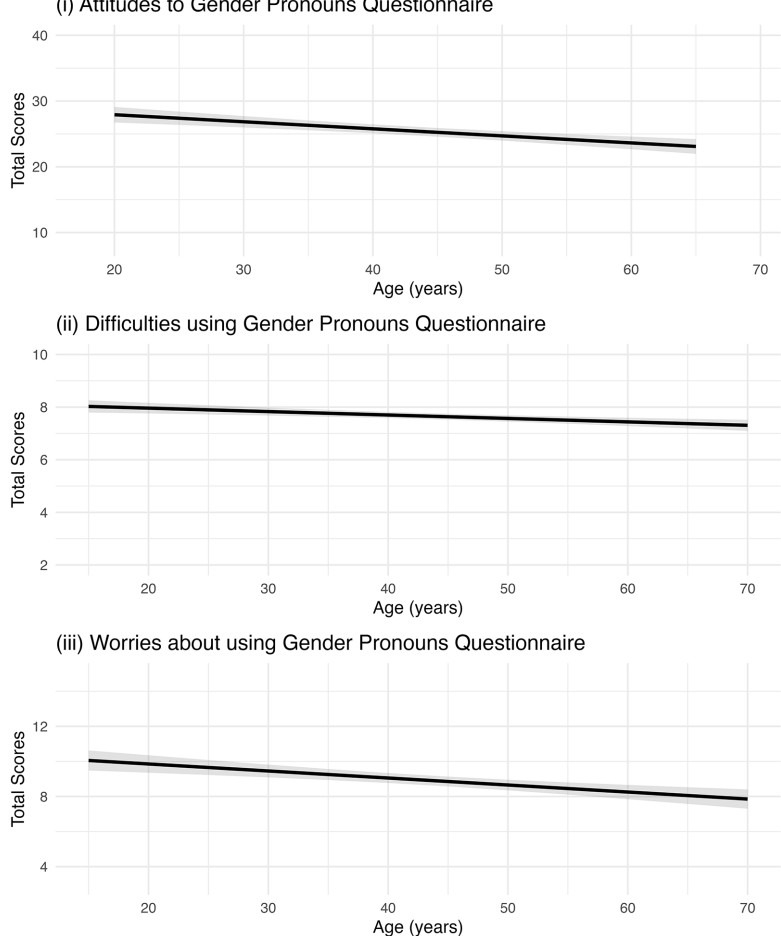

**Fig 2. Results from linear regression models for Hypothesis 2 showing predicted scores for (i) Attitudes to Gender Pronouns, (ii) Difficulties using Gender Pronouns, and (iii) Worries about using Gender pronouns in relation to age (years).** Shaded areas represent 95% confidence intervals based on robust variance estimates.

also reported greater difficulties using gender pronouns as indicated by lower scores ($b=-0.01$, 95% CI: $-0.02$, $-0.01$, $p<.001$; Table 3; Fig 2). We also explored the relationship between age and worries about using gender pronouns. Older participants reported less worries about using gender pronouns as indicated by lower scores ($b=-0.04$, 95% CI: $-0.06$, $-0.02$, $p<.001$; Table 3; Fig 2). This effect was weakened slightly but remained after adjusting for generalised anxiety (Table 3). Sensitivity analyses using proportional-odds ordinal models were consistent with these findings (S10 Table in S1 Appendix).

**Hypothesis 3: We hypothesise that self-reported attitudes towards gender pronouns and difficulties using gender pronouns will not be associated with performance on the pronoun-face task**

Consistent with Hypothesis 3, we did not find evidence that any of the attitude measures showed a statistically detectable association with interference scores (the difference in reaction times (ms) between incongruent and congruent trials), nor did these effects vary with age (all $ps>0.187$; Table 4). We also explored whether self-reported worries about using gender pronouns was associated with performance on the pronoun-face task. We did not find evidence that worries were associated with performance on the task, or that this varied by age (Table 4).

**Table 4. Results from linear regression models predicting interference effects (the difference in reaction times (ms) between incongruent and congruent trials) in the pronoun–face task from self-reported attitudes toward gender pronouns (overall attitudes, perceived difficulties, and worries), in interaction with age (years).**

| Predictor | b | Lower CI | Upper CI | p-value |
|---|---|---|---|---|
| *Overall Attitudes to Gender Pronouns* | | | | |
| Intercept | 112.08 | 47.50 | 176.65 | 0.001 |
| Sum Score | −0.98 | −3.30 | 1.33 | 0.405 |
| Age | −0.68 | −2.01 | 0.66 | 0.323 |
| Sum Score * Age | 0.03 | −0.02 | 0.08 | 0.187 |
| *Difficulties using Gender Pronouns* | | | | |
| Intercept | 127.83 | 19.41 | 236.26 | 0.021 |
| Sum Score | −5.16 | −18.90 | 8.58 | 0.461 |
| Age | −0.83 | −3.16 | 1.50 | 0.484 |
| Sum Score * Age | 0.13 | −0.17 | 0.43 | 0.409 |
| *Worries about using Gender Pronouns* | | | | |
| Intercept | 62.16 | 7.43 | 116.88 | 0.026 |
| Sum Score | 2.71 | −2.81 | 8.23 | 0.336 |
| Age | 0.54 | −0.59 | 1.67 | 0.347 |
| Sum Score * Age | −0.04 | −0.16 | 0.08 | 0.481 |

## Generational effects

We explored whether changes in processing of gender pronouns may be better characterised by differences in social experiences unique to generations by repeating our analyses for Hypotheses 1 and 2 with generation (Gen Z, Millennial, Gen X, Boomer) as a predictor in place of age. As findings were consistent, we report these results as supplementary (S6–S9 Tables in S1 Appendix).

## Discussion

This study investigated reaction times on a pronoun–face congruence task and questionnaire self-report measures of attitudes to pronoun gender categorization across a wide age range of adult participants. We had three main hypotheses. First, we predicted that we would find an age-related distinction between how generations performed on the pronoun–face task. Our second hypothesis was that that subjective experience would be different across age, with older participants reporting more negative attitudes as well as difficulty using conflicting or non-binary pronouns. Finally, we looked at explicit attitudes to pronoun assignment using regression modelling to determine whether explicit self-reports predicted pronoun–face interference effects. Reaction time responses and explicit measures often diverge [25], such as commonly found in the Implicit Attitudes Test [15]. As our initial pilot study indicated that there would be no relationship, we predicted that there would be no correlation between reaction time responses and explicit attitudes. We discuss our results in turn.

Critically, we did not find age differences in the pronoun–face congruence test, finding that all participants showed a similar expected pattern of quicker response times for congruent compared to incongruent trials. This is the finding we had observed in the initial pilot study that treated age as a categorical variable. Both analysis by age as a category or as a continuous variable in the current study did not find any age-related changes in the pronoun–face congruence task. As age may reflect both life-course variation as well as cohort differences in social experience, and these cannot be disentangled in a cross-sectional design, the convergence of findings across both approaches suggests that neither aging nor cohort effects meaningfully influenced performance on the task. All participants had quicker responses in the pronoun–face task compared to the control (orientation) condition. Quicker responses typically indicate greater processing fluency,

suggesting that pronoun–face judgments are cognitively simpler or more familiar than the orientation judgments. Additionally, task-related factors, such as the differences in cue length (e.g., 'upright' or 'upside down') versus pronoun cues (e.g., '*he*' or '*she*') or the unfamiliarity of inverted faces, may have contributed to the slower responses observed in the orientation task. We observed a general slowing of response times in older participants but not an increase in interference as in prior literature [26]. For example, Mutter, Naylor & Patterson [27] found that age related differences in Stroop tasks which also trigger interference are magnified in certain conditions (e.g., ones that promote transient failures to maintain task content) and it is possible that our task required less conceptual level inhibitory control than classic colour interference tests [28]. Also, while our pronoun–face task and the standard Stroop task both address interference, our congruence measure of match or mismatch is different to the standard Stroop measure which measures interference across a separate dimension.

In line with our second prediction, on explicit measures we found that older individuals reported more negative attitudes as well as difficulties about using non-binary pronouns. Additionally, age was associated with a decline in worries about using gender pronouns, even after adjusting for general levels of anxiety, suggesting that older participants may care less about the issue. This is consistent with literature characterising younger adults from Western societies as being socially conscious about gender identity [29,30], and presenting with increased self-awareness and acceptance of people and wider societal concerns [31,32]. A 2025 literature review [3] of attitudes to using gendered language found that "older individuals seem to find it easier than younger ones to recognize the negative impact of gendered language in everyday life *but* (emphasis added) were less inclined to adopt newly introduced forms".

Older generations from Western societies have more exposure to typical pronoun-face matches, stereotypical gender roles, and less experience with gender fluid peers. It would be difficult to tease apart the role of experience as it confounds so neatly with age. Our investigations using regression modelling did not discover any meaningful patterns between performance on the pronoun–face task and self-reports related to pronoun assignment.

This study demonstrates that binary pronouns trigger an automatic categorization processing based on biological sex distinction, and that interference was observed when the pronoun did not match a normative gender face. This aligns with literature on automatic categorisation of gender [33]. These social categories appear automatic with signatures of face processing even identifiable at neural levels. Rekow and colleagues [34] used EEG to demonstrate that gender categorisation in adults can occur automatically after a single glance at a picture, suggesting that this process is both rapid and hardwired. More recently Serafini & Pesciarelli [9] specifically looked at congruence between gendered words in Spanish that primed both target pronouns and faces. They found facilitated reaction times (RTs) and differential event-related potentials (ERPs) to a target third-person singular pronoun (*lui* 'he' or *lei* 'she') or face (male, female), preceded by grammatically marked or stereotypically associated words.

Although the cultural shift and societal transformation of gender conceptions [35] appears to affect subjective pronoun use and understanding, future work is needed to document whether such effects are widespread and how they manifest in actual incidences of misgendering and actual use of non-binary pronouns in live settings. We note that we are unable to make any claims about non-binary presentation or preferred non-binary pronoun use with the current interference methodology. Our study instead provides initial evidence for the automatic process supporting binary pronoun categorisation and provides a foundation for future work investigating the cognitive and social aspects of misgendering. Overall, our key finding is that despite subjective differences that might underpin behaviour, gender-based associations remain fast and implicit.

## Supporting information

**S1 Appendix. Supplementary analyses, pilot study results, questionnaire development, and additional mixed-effects and regression outputs.** This appendix contains all supplementary tables and the supplementary figure referenced in the manuscript.
(DOCX)

## Acknowledgments

We would like to sincerely thank April H. Bailey, Marie Gustafsson Sendén, Justin B. Hopkins, Lucille Kerr, Francisco Perales and Rieka von der Warth for their valuable input and feedback during initial item refinement for the Explicit Attitudes to Gender Pronouns Questionnaire.

## Author contributions

**Conceptualization:** Melissa Allen, Will Mills, Stephen Hinde, Bruce Hood.

**Data curation:** Catherine Hobbs, Will Mills, Stephen Hinde.

**Formal analysis:** Catherine Hobbs, Will Mills, Stephen Hinde.

**Investigation:** Melissa Allen, Catherine Hobbs, Bruce Hood.

**Methodology:** Melissa Allen, Catherine Hobbs, Stephen Hinde, Bruce Hood.

**Project administration:** Catherine Hobbs, Will Mills.

**Resources:** Will Mills, Stephen Hinde.

**Software:** Catherine Hobbs, Will Mills, Stephen Hinde.

**Supervision:** Melissa Allen, Bruce Hood.

**Visualization:** Catherine Hobbs.

**Writing – original draft:** Melissa Allen, Catherine Hobbs, Bruce Hood.

**Writing – review & editing:** Melissa Allen, Catherine Hobbs, Bruce Hood.

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
