## [Decision Letter · Decision Letter 0]

7 Oct 2025

Dear Dr. Hobbs,

Thank you for submitting your manuscript to PLOS ONE. After careful consideration, we feel that it has merit but does not fully meet PLOS ONE’s publication criteria as it currently stands. Therefore, we invite you to submit a revised version of the manuscript that addresses the points raised during the review process.

We look forward to receiving your revised manuscript.

Kind regards,

Christina M. Roberts

Academic Editor

PLOS ONE

Journal Requirements:

Reviewers' comments:

Reviewer's Responses to Questions

**Comments to the Author**

1. Is the manuscript technically sound, and do the data support the conclusions?

Reviewer #1: Partly

Reviewer #2: Yes

2. Has the statistical analysis been performed appropriately and rigorously?

Reviewer #1: Yes

Reviewer #2: No

3. Have the authors made all data underlying the findings in their manuscript fully available?

Reviewer #1: Yes

Reviewer #2: Yes

4. Is the manuscript presented in an intelligible fashion and written in standard English?

Reviewer #1: Yes

Reviewer #2: Yes

Reviewer #1: The manuscript addresses a highly interesting and socially relevant topic. The study is innovatively designed, and it is evident that the authors have made a commendable effort to ensure transparency and rigor in terms of materials, data collection, and analysis. However, the current version of the manuscript does not sufficiently explain key concepts, and it remains unclear what the experimental task is intended to measure and to what extent the experimental and explicit measures address overlapping concepts. As a result, the overall contribution of this study remains unclear. Specifically, the manuscript would benefit from a more precise theoretical framing of the experimental task and a clearer explanation of how attitudes are expected to manifest in the response patterns.

Specific comments:

The term 'gender pronoun attitude' (a key concept in the manuscript) is not clearly defined in the introduction. Furthermore, this concept is referred to inconsistently throughout the text (e.g. 'attitudes towards pronoun assignment' in the abstract, 'explicit attitudes to gender pronoun categorisation' on page 4, and 'self-perception of difficulty using pronouns' on page 5). A more precise and consistent definition is required to clarify the theoretical framework and the study's objectives.

I am not convinced that describing the task as a Stroop task is the best approach. The task appears to be more accurately described as a pronoun–face congruence task, in which participants judge the semantic match or mismatch between a face and a pronoun. That is, they must decide whether the gender association evoked by the face matches that evoked by the pronoun. In a Stroop-like task, I would have expected automatic gender associations to interfere with the processing of another dimension, such as categorising photos presented visually, regardless of the gender of the pronouns presented, where inhibition and control mechanisms would be relevant.

I am also unsure what the task is intended to measure. In terms of attitude, one could argue that the quicker someone can identify congruent trials (e.g. a photo of a woman paired with the word 'she') as matches and incongruent trials (e.g. a photo of a man paired with the word 'she') as mismatches, the more traditional their gender assignments tend to be.

As gender-congruent pairs are expected to be more familiar, differences in recognition times between matches and mismatches could reflect processing costs, as familiar combinations are easier to process. However, I am unsure how this can be interpreted as an expression of implicit 'attitudes towards gender pronouns'.

It would be helpful if the authors would provide a more detailed explanation of the cognitive processes involved in performing the task they designed, along with the rationale behind it

The extent to which the experimental task and the explicit measures (questionnaire) address overlapping constructs is unclear. The explicit measure assesses attitudes towards the assignment of pronouns based on biological characteristics versus self-identification. However, it is unclear how a positive attitude towards self-selected pronouns would manifest in the response patterns of the experimental task. For example, if a participant believes that pronouns should not be assigned based on biological or facial features, they might consider a traditionally masculine-looking face paired with 'she' to be an appropriate match. However, such responses appear to have been excluded from the analysis as 'inaccurate'. It would be important to explicitly explain how attitudes are expected to manifest in the response patterns.

Reviewer #2: In general, I liked the idea of the study, especially the fact of including the age factor to be analyzed. Still, I do not think that the paradigm is very novel (other authors have used very similar tasks), the literature review is not very actual, and I miss some conditions to deeply explore topics related to alternative gender pronouns and gender fluid representation. As a positive note, I would like to remark on the effort of the authors in collecting a huge amount of data, and on doing a very sophisticated pilot study. This provides great statistical support to the findings, which is laudable.

Below I present chronically my comments and suggestions, which I think will increase the quality of the study.

.

Reviewer #1: No

Reviewer #2: **Yes:**Alba CasadoAlba CasadoAlba CasadoAlba Casado

---

## [Author Response · Author response to Decision Letter 1]

14 Nov 2025

Dear Dr Roberts,

Thank you for the opportunity to revise and resubmit our manuscript, previously titled “Age-Related Discrepancies in Gender Pronoun Attitudes: a Mismatch between Implicit and Explicit Measures” for consideration for publication in PLOS One. In line with Reviewer 1’s comments regarding the framing of the paper, we have now re-titled it “My Pronouns Are”: An Investigation of Pronoun-Face Mismatch and Self-Report Attitudes to Gender Categorizaton Across Generations”. We have changed the name of the task from gender pronoun Stroop task to pronoun-face congruence task to reflect the reviewers’ valid point that our task is not a classic Stroop paradigm. We have also gone through all supplementary materials and figures and worded accordingly. We appreciate the detailed feedback from the reviewers and outline how we have responded to each point below. We feel these revisions have deeply strengthened the paper and thank you and the reviewers for your engagement and helpful suggestions.

Reviewer #1: The manuscript addresses a highly interesting and socially relevant topic. The study is innovatively designed, and it is evident that the authors have made a commendable effort to ensure transparency and rigor in terms of materials, data collection, and analysis. However, the current version of the manuscript does not sufficiently explain key concepts, and it remains unclear what the experimental task is intended to measure and to what extent the experimental and explicit measures address overlapping concepts. As a result, the overall contribution of this study remains unclear. Specifically, the manuscript would benefit from a more precise theoretical framing of the experimental task and a clearer explanation of how attitudes are expected to manifest in the response patterns.

Specific comments:

The term 'gender pronoun attitude' (a key concept in the manuscript) is not clearly defined in the introduction. Furthermore, this concept is referred to inconsistently throughout the text (e.g. 'attitudes towards pronoun assignment' in the abstract, 'explicit attitudes to gender pronoun categorisation' on page 4, and 'self-perception of difficulty using pronouns' on page 5). A more precise and consistent definition is required to clarify the theoretical framework and the study's objectives.

Response: We have defined and been more consistent with our terminology. We now use the term explicit attitudes to preferred gender pronouns and pronoun categorization and define this as beliefs regarding traditional and preferred pronouns. This is explored here using our novel questionnaire which investigates attitude, difficulty and worry using gendered pronouns.

I am not convinced that describing the task as a Stroop task is the best approach. The task appears to be more accurately described as a pronoun–face congruence task, in which participants judge the semantic match or mismatch between a face and a pronoun. That is, they must decide whether the gender association evoked by the face matches that evoked by the pronoun. In a Stroop-like task, I would have expected automatic gender associations to interfere with the processing of another dimension, such as categorising photos presented visually, regardless of the gender of the pronouns presented, where inhibition and control mechanisms would be relevant.

Response: We thank the reviewer for bringing this to our attention and agree that the paradigm is better described as a pronoun-face congruence task with a match/mismatch response which differs from the classic Stroop paradigm; we have altered the manuscript accordingly throughout. We also further describe the origin of our task in relation to the different auditory tasks (Green & Barber, 1981; Most et al, 2007) and explain why we were looking at congruence as a measure of interference. We have removed discussion of the longitudinal Stroop studies and refer to more recent work looking at interference in general. We also acknowledge that we do not know whether it is familiarity or interference that generates our pronoun-face congruence interference and that it has not been studied longitudinally. We also acknowledge in discussion that the classic Stroop generates interference across another dimension.

I am also unsure what the task is intended to measure. In terms of attitude, one could argue that the quicker someone can identify congruent trials (e.g. a photo of a woman paired with the word 'she') as matches and incongruent trials (e.g. a photo of a man paired with the word 'she') as mismatches, the more traditional their gender assignments tend to be. As gender-congruent pairs are expected to be more familiar, differences in recognition times between matches and mismatches could reflect processing costs, as familiar combinations are easier to process. However, I am unsure how this can be interpreted as an expression of implicit 'attitudes towards gender pronouns'. It would be helpful if the authors would provide a more detailed explanation of the cognitive processes involved in performing the task they designed, along with the rationale behind it.

Response: We agree with the reviewer’s summation that speeded responses on our pronoun-face congruence task is likely to reflect familiarity (which we have specifically addressed further in the introduction), We have removed all referral to implicit attitudes throughout and simply talk about congruence/incongruence interference. We have further elaborated why we were looking at processing costs based on the Implicit Attitudes Test literature but also include discussion of the controversial interpretation of how it related to explicit attitudes (see same point below).

The extent to which the experimental task and the explicit measures (questionnaire) address overlapping constructs is unclear. The explicit measure assesses attitudes towards the assignment of pronouns based on biological characteristics versus self-identification. However, it is unclear how a positive attitude towards self-selected pronouns would manifest in the response patterns of the experimental task. For example, if a participant believes that pronouns should not be assigned based on biological or facial features, they might consider a traditionally masculine-looking face paired with 'she' to be an appropriate match. However, such responses appear to have been excluded from the analysis as 'inaccurate'. It would be important to explicitly explain how attitudes are expected to manifest in the response patterns.

Response: First, we agree with the reviewer that the relationship between reaction time interference tasks and explicit attitudes is not straightforward (as per the controversial over interpretation of the Implicit Attitudes Test that we address in the introduction). We have sought to clarify why one might expect a relationship in the introduction. Specifically, individuals who explicitly expressed difficulty in applying pronouns might have had stronger biases towards typical match/mismatch categorization on a speeded task. However, as in our earlier pilot, this prediction was not supported here, even after improving the explicit measure (now validated) and increasing the sample to address the earlier power concern. We now state this more clearly in our discussion.

As to the second point, it is true that if an individual consistently believed that pronoun-face incongruence to be an appropriate match, then they would have been removed from analysis as this would have resulted in low accuracy. Our pre-registered data-quality criterion (< 70% accuracy on ≥1 task) follows standard practice for speeded RT paradigms, where very low accuracy typically indicates that RTs are not interpretable (e.g. guessing, inattention). Twenty-six participants met this preregistered exclusion. As the reviewer notes, it is possible that a small subset of participants responded “inaccurately” because they personally endorsed pronoun–face pairings that differed from the experimental rule. We explored this by looking for the pattern one would expect under that interpretation, namely relatively good performance on the orientation task but poor performance on the pronoun-face task. We found only 9 such cases (≈1.3% of participants who completed the study procedure). This number is too small to re-estimate the RT–attitude relationship with adequate power, so we have not conducted a separate analysis. However, to facilitate further work on this question, we have made the full dataset, including excluded cases, openly available so that alternative scoring rules or inclusion criteria can be applied.

Reviewer #2:

In general, I liked the idea of the study, especially the fact of including the age factor to be analyzed. Still, I do not think that the paradigm is very novel (other authors have used very similar tasks), the literature review is not very actual, and I miss some conditions to deeply explore topics related to alternative gender pronouns and gender fluid representation. As a positive note, I would like to remark on the effort of the authors in collecting a huge amount of data, and on doing a very sophisticated pilot study. This provides great statistical support to the findings, which is laudable.

Response: We appreciate that the reviewer likes the idea, the age analysis and the great statistical support. We have reframed the paper in line with Reviewer 1’s suggestion above and reworked the literature review accordingly.

The study is indeed similar to other older auditory Stroop studies (e.g. Green & Barber (1981) Auditory Stroop. Male v Female Voice speaking “Man” v “Girl” Most et al., (2007) uses gender category words and vocal response “Boy” or “Girl” Looked at children and adults). However, as suggested, we have added the most recent study by Serafini, L., & Pesciarelli, F. (2025) which looked at whether gendered words primed Italian he/she pronouns or male/female faces. We agree that our study cannot deeply explore topics related to alternative gender pronouns gender fluid representation, and acknowledge this in the discussion.

Below I present chronically my comments and suggestions, which I think will increase the quality of the study.

Pg. 9 line 36 “a uniquely millennial phenomenon which sees many young people rejecting the notion that sex and gender need to be correlated, with some individuals wholly rejecting gender as a binary construct”

– I do not think this is an exclusive millennial phenomenon. Indeed, many cultures have gender fluid members well categorized for centuries. See for instance Hijra (South Asia), Muxe (Oxaca, Mexico), Bugis, Bissu, Calalai (Indonesia), Mahu (Hawaii) among others. This has been a reality for non-western societies, and we should not invalidate them as individuals or talk about it as a fashion for young people. People who do not categorize their gender with their biological sex has always existed and we should acknowledge this reality.

Response: We thank the reviewer for this important point. We have removed the phrasing ‘uniquely millennial’ and specified this is in relation to Western cultures. We have also now explicitly acknowledged the range and reality of gender fluid cultures at the end of the introduction section.

Pg. 9 line 39 “coding for sex is automatic and difficult to suppress in adults”

– the reference for this quote is a study about race, not sex. Moreover, the result of the study posits that it is possible to change the perception of race: “an alternate social world was enough to deflate the tendency to categorize by race. ” Therefore, I think that this statement needs to be supported by the literature, or either toned down because despite the coding might be automatic, I am not sure it will be hard to suppress.

Response: We have removed the reference to race and provided an alternative source related to gender categorization.

Pg. 9 line 40: cite no. 4 reference a study from 1978. I suggest doing another search in the literature so you could include more recent studies tackling the same topic. For instance,it comes to my mind the study by Van den Brink et al., 2012 with voice processing that is also rather old. For sure, more recent research has been done, please try to find something newer.

Response: We have included more recent research as requested (eg. Serafini, L., & Pesciarelli, F. 2025, Reference No. 9).

Pg. 9 line 53: cite no. 6: the study is also a bit old. You can find more recent references and similar studies using the sex of the speaker to trigger incongruence, like for instance the study by Casado et al., 2021

Response: We have now included the Casado et al. (2021) reference and thank the reviewer for pointing us to this paper. We agree that the other stated citation is a bit old, but have chosen to keep it in the paper as it is the most directly relevant to our paradigm because it looks at voices and pronouns that are congruent and incongruent and complements the Casado, Palma & Paolieri paper.

Pg. 10 line 59: cite no. 7 from 1990. Include also a reference more contemporary.

Response: We have included another more recent reference (Pettigrew et al., 2014) that shows interference increases with age.

Pg 10 line 61: cite no. 8 – I am sure that a more recent metanalysis has been done since

1998, please, do a literature search to actualize the reference.

Response: In accordance with the request of Reviewer 1, we have removed this meta-analysis and much of the discussion related to Stroop tasks.

Pg 10 line 63 “As individuals gain experience with certain types of faces, especially those encountered more frequently, facial processing becomes more efficient but also more entrenched”

– this information is rather surprising for the reader. Until now you are talking

about interference and inhibition in general, and suddenly there is a jump to face

processing. I miss a link of information here.

Response: We have added another example (the other race effect; Hancock & Rhodes, 2008; Reference No. 13) of face processing which displays this entrenchment phenomenon related to experience to draw the link between strength of pronoun-face matching with age.

Pg. 11 line 81: “In this study…”

– I think your procedure is very similar to the one used by

Serafini & Pesciarelli 2025. Please, have a look at this study and cite them.

Response: This study is now cited earlier (Reference No. 9).

Pg. 18 line 222 “Participant was included as random intercept to account for within-

subject variability”

– I think this is the correct decision. However, I wonder why you did not include the faces as random intercept as well, this way, you could also account to the variability of your stimuli and will increase the power of your analyses. I strongly suggest the Authors to include the stimuli (item) intercept, and also, to create the maximal models following Barr et al., 2013 recommendation. In your case, (1 + stroop task + congruency| participants) + (1 + age + stroop task + congruency | items). Aso, make sure before including the continuous variables in the task that they are normally distributed (including RTs), otherwise you may need a transformation. Moreover, specify the contrast selected for the categorical variables—if you do not have specific predictions about the direction, Schad et al. (2020) usually recommend using a sum contrast (-0.5, 0.5)

Response: Thank you for the suggestion to consider the effect of faces. We did not do this in our original analysis as we modelled data at a mean RT level (i.e., collapsed across trials). We have now addressed this by conducting an additional trial-level analysis which takes into account the effect of faces, in line with the reviewer’s suggestions.

As the reviewer suggested, we initially used a maximal model following Barr et al’s recommendations (2013): log_RT ~ age * task * congruency + (1 + task * congruency || participant) + (1 + task * congruency || face). However, this model failed to converge and produced singular fits, indicating that several variance components, particularly the by-face slopes and the by-participant task slope, were effectively zero and not supported by the data. We would like to note that we did not include age as a random effect in this model as age is a betw

---

## [Decision Letter · Decision Letter 1]

11 Dec 2025

Dear Dr. Hobbs,

Thank you for submitting your manuscript to PLOS ONE. After careful consideration, we feel that it has merit but does not fully meet PLOS ONE’s publication criteria as it currently stands. Therefore, we invite you to submit a revised version of the manuscript that addresses the points raised by Reviewer 1 during the review of your revised manuscript.

We look forward to receiving your revised manuscript.

Kind regards,

Christina M. Roberts, M.D., M.P.H.

Academic Editor

PLOS One

Journal Requirements:

Reviewers' comments:

Reviewer's Responses to Questions

**Comments to the Author**

Reviewer #1: (No Response)

Reviewer #2: All comments have been addressed

2. Is the manuscript technically sound, and do the data support the conclusions?

Reviewer #1: Yes

Reviewer #2: Yes

3. Has the statistical analysis been performed appropriately and rigorously?

Reviewer #1: Yes

Reviewer #2: Yes

4. Have the authors made all data underlying the findings in their manuscript fully available?

Reviewer #1: Yes

Reviewer #2: Yes

5. Is the manuscript presented in an intelligible fashion and written in standard English?

Reviewer #1: Yes

Reviewer #2: Yes

Reviewer #1: Overall, the manuscript shows significant improvement and much greater clarity. Most of my previous comments have been satisfactorily addressed. One issue remains unresolved, and I have a few additional comments for consideration. Thank you for your careful revisions and for engaging thoughtfully with the feedback—I appreciate the effort invested in strengthening the manuscript.

There remains a lack of conceptual clarity regarding the self-report measure. In the supplementary materials, the questionnaire is presented as “Attitudes towards gender pronouns” with content validity checked by academic experts. Including the instructions provided to these experts in the main text might help clarify the underlying concept(s).

The phrase in line 98f—“Attitudes towards gender pronouns categorisation (e.g., beliefs regarding traditional and preferred pronouns)”—does not sufficiently explain the construct being measured. A precise definition or detailed description is needed to specify the attitude object. For example, what exactly is meant by “gender pronouns categorisation”?

As you intend to present your study to an international audience, it would be helpful to explicitly state what is meant by “traditional” and “preferred” pronouns in English, as these terms may vary across languages.

Additionally, the abstract refers to “attitudes towards misgendering.” Are these the same attitudes or distinct constructs? Please clarify and, if they differ, define both explicitly and ensure consistent terminology throughout the manuscript.

Looking at the items in Table 1 (Attitudes to gender pronouns), some appear to assess whether participants view pronouns as strictly binary (he/she) or accept non-binary options, others reflect attitudes toward the social norm of sharing pronouns, and others relate to respecting individual identity. Please note that even if a scale demonstrates internal reliability (i.e., consistent responses across items), its items can still cover multiple themes. This should be acknowledged and conceptually addressed.

In sum, the introduction should include a clear and comprehensive definition of the attitude construct. This definition could also incorporate the two sub-scales—difficulties in using gender pronouns and worries about using gender pronouns—as they seem to conceptually belong to the overarching attitude construct, even though they were empirically distinguished through factor analysis. Including this clarification would help readers understand how the sub-scales relate to the main construct and strengthen the conceptual framework of the measure.

The manuscript suggests both age-related changes (e.g., cognitive slowing and changes in inhibitory control) and cohort effects (e.g., differences in social norms across generations), with age-related changes linked to interference effects in the congruence tasks and cohort effects linked to the self-report measure. Currently, these perspectives appear in different parts of the manuscript. I recommend making this contrast more explicit, especially since testing age as both a continuous and categorical variable is informative but does not fully disentangle age-related changes from cohort effects. For example, this distinction could be highlighted when presenting the hypotheses to clarify which effects are expected to reflect aging and which reflect generational differences.

I consider the mixed-effects modeling approach that includes faces as a random effect when testing Hypothesis 1 as superior, as this method avoids the loss of information caused by averaging across trials. Additionally, the table in the supplementary materials indicates a congruency-by-task interaction with p < .20. Based on Figure 1, this appears to suggest that the interference effect is slightly stronger in the orientation task. This interpretation would be consistent with the generally higher response times observed for the orientation-face task, which likely reflects greater overall task difficulty. While neither analytical approach provides support for Hypothesis 1, I suggest considering whether this small difference in findings might still be relevant to report for completeness.

Minor:

Caution is warranted with reference to the findings for Hypothesis 4, as the interaction between Overall Attitudes to Gender Pronouns and Age seems to yield a p-value below .20. This result should be reported. Since the hypothesis predicted a null finding, note that the result is consistent with this expectation (as it does not meet convential significance criteria) but should not be interpreted as proof of the null.

When discussing the faster responses to the pronoun–face task compared to the orientation task (line 375ff), the explanation appears confusing, as faster responses are typically interpreted as indicating more automatic processing. I recommend clarifying this point and, additionally, considering other factors that may contribute to the observed difference. For example, task complexity—words in the orientation task are longer—and stimulus familiarity—participants may be more familiar with up–down photos than down–up photos. Including these considerations would provide a more comprehensive interpretation of the response time differences.

Please ensure that all labels used in figures and tables are clearly explained in the captions or text and that they are consistent with the terminology used throughout the manuscript (for example, the label for the criterion used in the analysis presented in Table 4 - interference effect - should have been introduced when presenting the statistical analysis for the respective hypothesis (l. 257).

Ensure that the table headers for regression analyses clearly state the criterion and the predictors; for example the header in Table 4 would be easier to comprehend as “Linear regression models predicting interference effects in the pronoun–face task from self-reported attitudes toward gender pronouns (overall attitudes, perceived difficulties, and worries), in interaction with age (years)”.

Consider standardising the way hypotheses are presented. In academic writing, it is common to use numerals and to capitalise the term; for example: ‘Hypothesis 1’ rather than ‘hypothesis one.’ (also ‘Dataset 1’ rather than ‘dataset 1’)

Reviewer #2: Thanks for addressing all my comments and suggestions. I believe that with the current structure the paper is ready for publication

.

Reviewer #1: No

Reviewer #2: **Yes:**Alba CasadoAlba CasadoAlba CasadoAlba Casado

---

## [Author Response · Author response to Decision Letter 2]

9 Jan 2026

Reviewer #1: Overall, the manuscript shows significant improvement and much greater clarity. Most of my previous comments have been satisfactorily addressed. One issue remains unresolved, and I have a few additional comments for consideration. Thank you for your careful revisions and for engaging thoughtfully with the feedback—I appreciate the effort invested in strengthening the manuscript.

Response: We thank the reviewer for their careful reading of the revised manuscript and for the constructive feedback throughout the review process. We appreciate the detailed engagement with our work and the acknowledgement of the improvements made. We address the remaining issue and all additional comments in full below and have made the corresponding revisions in the manuscript.

There remains a lack of conceptual clarity regarding the self-report measure. In the supplementary materials, the questionnaire is presented as “Attitudes towards gender pronouns” with content validity checked by academic experts. Including the instructions provided to these experts in the main text might help clarify the underlying concept(s).

Response: We are currently drafting a research paper fully detailing the construction and psychometric validation of the self-report measure we have used in this study. We believe the full methods and results of this process would be beyond the remit of the current research paper. However, we have detailed the process briefly in the supplementary materials to give sufficient background context for the use of the self-report measure in this study. In response to the reviewer’s suggestion, we have now clarified the instructions given to the academic experts in the supplementary materials.

The phrase in line 98f—“Attitudes towards gender pronouns categorisation (e.g., beliefs regarding traditional and preferred pronouns)”—does not sufficiently explain the construct being measured. A precise definition or detailed description is needed to specify the attitude object. For example, what exactly is meant by “gender pronouns categorisation”?

As you intend to present your study to an international audience, it would be helpful to explicitly state what is meant by “traditional” and “preferred” pronouns in English, as these terms may vary across languages.

Response: In the introduction as requested, we have added clarification of the process we call gender pronoun categorization which we regard is the normative pattern of pronoun assignment that is typically used in various contexts. One such context is the assignment of gender pronouns based on physical appearance of an individual which is the focus of the current set of studies. However, we have also acknowledged that these categories are not fixed, universal nor do they capture the full diversity that exists beyond our paradigm.

Additionally, the abstract refers to “attitudes towards misgendering.” Are these the same attitudes or distinct constructs? Please clarify and, if they differ, define both explicitly and ensure consistent terminology throughout the manuscript.

Response: We have amended this to refer to attitudes toward gender pronoun categorization.

Looking at the items in Table 1 (Attitudes to gender pronouns), some appear to assess whether participants view pronouns as strictly binary (he/she) or accept non-binary options, others reflect attitudes toward the social norm of sharing pronouns, and others relate to respecting individual identity. Please note that even if a scale demonstrates internal reliability (i.e., consistent responses across items), its items can still cover multiple themes. This should be acknowledged and conceptually addressed.

Response: Our EFA and CFA supported a one-factor solution with good fit, indicating that the items share substantial common variance and can be treated as a unidimensional scale. However, in line with the reviewer’s suggestion we recognise that whilst the item constructs are strongly correlated in our sample, they are conceptually distinguishable. We now clarify this in the manuscript and note that the total score reflects a broad attitudinal dimension encompassing these related components.

In sum, the introduction should include a clear and comprehensive definition of the attitude construct. This definition could also incorporate the two sub-scales—difficulties in using gender pronouns and worries about using gender pronouns—as they seem to conceptually belong to the overarching attitude construct, even though they were empirically distinguished through factor analysis. Including this clarification would help readers understand how the sub-scales relate to the main construct and strengthen the conceptual framework of the measure.

Response: We thank the reviewer for this helpful suggestion. We have revised the Introduction to provide a clearer conceptual definition of the self-report measures and their relationships. Specifically, we now clarify that overall attitudes toward gender pronouns were assessed using a validated attitudinal scale, while perceived difficulties using pronouns and worries about misgendering were assessed using separate self-report scales. We explain that difficulties reflect perceived cognitive or linguistic challenges in pronoun use, whereas worries capture affective concerns about using pronouns incorrectly and causing social harm. Although these constructs were empirically distinguishable, we note that they are conceptually related in that they reflect how individuals evaluate, engage with, and emotionally respond to changing norms around gender categorization. This clarification has been added to the Introduction to strengthen the conceptual framework and aid reader interpretation.

The manuscript suggests both age-related changes (e.g., cognitive slowing and changes in inhibitory control) and cohort effects (e.g., differences in social norms across generations), with age-related changes linked to interference effects in the congruence tasks and cohort effects linked to the self-report measure. Currently, these perspectives appear in different parts of the manuscript. I recommend making this contrast more explicit, especially since testing age as both a continuous and categorical variable is informative but does not fully disentangle age-related changes from cohort effects. For example, this distinction could be highlighted when presenting the hypotheses to clarify which effects are expected to reflect aging and which reflect generational differences.

Response: Thank you for this insightful comment. We agree that age can reflect both life-course (aging-related) variation and cohort differences in social experience. We have now clarified this conceptual distinction in the Introduction by noting that age may capture both aging-related and generational influences. We also added a brief explanation in the Methods describing why we examined both continuous and categorical age predictors, and we now explicitly state in the Results that generational analyses yielded the same pattern as the continuous-age models. Finally, we have revised the Discussion to acknowledge that, because the study is cross-sectional, aging and cohort effects cannot be fully disentangled, and that the convergence of continuous-age and generational analyses suggests that neither type of effect meaningfully altered our conclusions.

I consider the mixed-effects modeling approach that includes faces as a random effect when testing Hypothesis 1 as superior, as this method avoids the loss of information caused by averaging across trials. Additionally, the table in the supplementary materials indicates a congruency-by-task interaction with p < .20. Based on Figure 1, this appears to suggest that the interference effect is slightly stronger in the orientation task. This interpretation would be consistent with the generally higher response times observed for the orientation-face task, which likely reflects greater overall task difficulty. While neither analytical approach provides support for Hypothesis 1, I suggest considering whether this small difference in findings might still be relevant to report for completeness.

Response: We thank the reviewer for their thoughtful observation. We agree that the mixed-effects analysis retains more trial-level information and therefore included this supplementary exploratory analysis for completeness. However, as this analysis was not part of our preregistered plan, we are cautious about interpreting patterns that may arise by chance, particularly given the increased Type I error risk associated with unplanned analyses.

Although the congruency × task interaction in the trial-level model had a numerical value in an interesting direction, its p-value (p = 0.178) and confidence interval both clearly encompassed zero, indicating no statistical evidence for an interaction. Consistent with this, the effect size was extremely small. Importantly, the study was highly powered to detect such small effects, we had a large sample (N = 600) and over 110,000 trial-level observations. The absence of a significant interaction in this context therefore suggests that any difference between tasks is negligible.

For these reasons, and to avoid overinterpretation of exploratory findings that are not statistically robust, we have chosen not to highlight the p = 0.178 effect in the main text, but we continue to report the full model results in the supplementary materials for transparency.

Minor:

Caution is warranted with reference to the findings for Hypothesis 4, as the interaction between Overall Attitudes to Gender Pronouns and Age seems to yield a p-value below .20. This result should be reported. Since the hypothesis predicted a null finding, note that the result is consistent with this expectation (as it does not meet convential significance criteria) but should not be interpreted as proof of the null.

Response: We thank the reviewer for this helpful clarification. In our preregistered analysis for Hypothesis 3 (which we believe the reviewer is referring to, with results presented in Table 4), the Attitudes × Age interaction (p = 0.187) and all related predictors had confidence intervals that clearly included zero, indicating no statistically detectable association with interference scores. Because our hypothesis predicted no effect, these non-significant results are consistent with this expectation; however, we agree that they should not be interpreted as evidence for the null, and we do not treat p-values above .05 as meaningful. We have amended the Results section to state that none of the attitude predictors nor their interactions with age showed a statistically detectable association with interference scores (all ps for predictors > .18), without implying that the absence of significance constitutes proof of the null.

When discussing the faster responses to the pronoun–face task compared to the orientation task (line 375ff), the explanation appears confusing, as faster responses are typically interpreted as indicating more automatic processing. I recommend clarifying this point and, additionally, considering other factors that may contribute to the observed difference. For example, task complexity—words in the orientation task are longer—and stimulus familiarity—participants may be more familiar with up–down photos than down–up photos. Including these considerations would provide a more comprehensive interpretation of the response time differences.

Response: Thank you for highlighting this. We agree that our original interpretation required clarification. We have revised the discussion to provide a clearer and more accurate explanation of why responses were faster in the pronoun–face task. Specifically, we now note that faster responses likely reflect greater processing fluency, and we additionally acknowledge that task-related factors may also contribute to the observed differences in response times. This revision offers a more comprehensive and balanced interpretation, in line with the reviewer’s suggestion.

Please ensure that all labels used in figures and tables are clearly explained in the captions or text and that they are consistent with the terminology used throughout the manuscript (for example, the label for the criterion used in the analysis presented in Table 4 - interference effect - should have been introduced when presenting the statistical analysis for the respective hypothesis (l. 257).

Response: We have now amended the methods and results section for Hypothesis 3 to indicate that the interference effect is calculated as the difference in reaction times (ms) between incongruent and congruent trials.

Ensure that the table headers for regression analyses clearly state the criterion and the predictors; for example the header in Table 4 would be easier to comprehend as “Linear regression models predicting interference effects in the pronoun–face task from self-reported attitudes toward gender pronouns (overall attitudes, perceived difficulties, and worries), in interaction with age (years)”.

Response: We have amended the titles for all of the tables in line with the reviewer’s suggestion.

Consider standardising the way hypotheses are presented. In academic writing, it is common to use numerals and to capitalise the term; for example: ‘Hypothesis 1’ rather than ‘hypothesis one.’ (also ‘Dataset 1’ rather than ‘dataset 1’)

Response: We have amended this.

Reviewer #2: Thanks for addressing all my comments and suggestions. I believe that with the current structure the paper is ready for publication

Response: We sincerely thank the reviewer for their thoughtful engagement with our manuscript and for the helpful comments and suggestions provided throughout the review process. We are very grateful for the time and care taken in evaluating our work, and we are pleased that the revised manuscript is now considered ready for publication.

---

## [Decision Letter · Decision Letter 2]

4 Feb 2026

“My Pronouns Are”: Pronoun-Face Mismatch Performance and Self-Report Attitudes to Gender Categorizaton Across Generations

PONE-D-25-31694R2

Dear Dr. Hobbs,

We’re pleased to inform you that your manuscript has been judged scientifically suitable for publication and will be formally accepted for publication once it meets all outstanding technical requirements.

Kind regards,

Christina M. Roberts, M.D., M.P.H.

Academic Editor

PLOS One

Additional Editor Comments (optional):

Thank you for your responses to our reviewer. I feel your responses have improved your manuscript and it is now suitable for publication in PLOS One. Congratulations.

Reviewers' comments:

Reviewer's Responses to Questions

**Comments to the Author**

Reviewer #1: All comments have been addressed

2. Is the manuscript technically sound, and do the data support the conclusions?

Reviewer #1: Yes

3. Has the statistical analysis been performed appropriately and rigorously?

Reviewer #1: Yes

4. Have the authors made all data underlying the findings in their manuscript fully available?

Reviewer #1: Yes

5. Is the manuscript presented in an intelligible fashion and written in standard English?

Reviewer #1: Yes

Reviewer #1: (No Response)

.

Reviewer #1: No

---

## [Editor Report · Acceptance letter]

PONE-D-25-31694R2

PLOS One

Dear Dr. Hobbs,

I'm pleased to inform you that your manuscript has been deemed suitable for publication in PLOS One. Congratulations! Your manuscript is now being handed over to our production team.

Kind regards,

on behalf of

Dr. Christina M. Roberts

Academic Editor

PLOS One